# SUPERMODEL: RETHINKING DNN TRAINING AND TESTING WITH OPEN-STYLE SKILL ACQUISITION AND DYNAMIC INFERENCE

## ABSTRACT

Current DNN model building suffers from two serious problems: forgetting and doomed test cases. In this paper, we propose an open-style skill acquisition approach, which is the opposite of a currently closed-style training scheme with recent features/patterns often overwriting previous ones to minimize the overall loss in backpropagation (the forgetting problem). Testing is also drastically different and is conducted as optimally selecting the best available skills (nodes and connections in DNNs) from the training model specific to a testing sample in order to maximize its probability to be correctly processed (the doomed test case problem). We validate our approach with multiple datasets and achieve significant performance improvement over SOTA methods.

**Keywords:** Machine Learning, Generalization, Deep Neural Network, Continual Learning, Lifelong Learning, Catastrophic Forgetting, Ensemble Modeling

## 1 INTRODUCTION

Modern DNN modeling consists of a training phase and a testing phase (Goodfellow et al., 2016). During training, first a fixed DNN architecture is selected based on data characteristics (e.g., modality, size), computation budget, and other factors such as incorporation of appropriate inductive bias. Once a model is selected, its architecture including number of layers and parameters will remain fixed through training and testing. Then a training set is divided into batches, and a loss is calculated after each batch, which will be used to update weights with backpropagation. When loss converges, training is finished, and the DNN model is fully built and passed to testing phase. The model will remain completely fixed during testing, and produce an evaluation measure with a testing set. In this paper, we will focus on the following two issues in current learning setting: forgetting of learned skills and doomed test cases.

**Forgetting of learned skills.** Forgetting of learned skills in human learning is usually not by choice, instead more due to aging, disease, and other factors. For DNNs, we use "forgetting" to denote that a set of model parameters — skill (formally defined in Methodology section) — that once were able to correctly classify a sample (either from a training set or testing set) are overwritten/updated during backpropagation, and can not correctly classify this sample any more. In the extreme case of catastrophic forgetting (Jagielski et al., 2022) (Kirkpatrick et al., 2017), when a neural network is trained with a sequence of tasks, weights learned in one task will be overwritten when updating the network to learn the next task. Besides catastrophic forgetting, there exists a subtle yet more common forgetting that has been reported in literature Toneva et al. (2019). In this case, forgetting occurs in standard batch-based DNN training for one task, and includes two scenarios. Firstly, within a batch forgetting occurs by choice (although a forced choice) because the overall training objective is to minimize the total loss of a whole batch. If one update of weights results in a larger difference between correctly predicted samples and incorrect ones, this update will be taken. The logic is that it is worth sacrificing a few previously correctly predicted samples to achieve a lower loss for the whole batch. The second scenario of forgetting occurs between batches, which is due to the usual choice of stochastic gradient descent (SGD) over gradient descent (GD) in current batch-based DNN training scheme . To speed up convergence and lower memory space requirement, batch-based SGD is the popular DNN choice over whole set-based GD. SGD usually can achieve a good approximation

of true (e.g., based on the whole training set) gradients, which makes it a reasonable choice for backpropagation. However, from the viewpoint of forgetting, SGD brings a serious problem. If we treat a neural network as a system of equations, weights $\{w_k\}$ will be variables in this system. Given a labeled sample $s = [x_1, x_2, ......, x_n, y]$, where $x_i$ is a feature, $y$ is the sample label, $s$ will become an equation in this system. DNN training becomes to find a solution for this system of equations. If each time we only take a subset of equations from this system, even if we can find a perfect solution $\{w_k\}$ for one subset/batch, this perfect solution will be (at least partially) overwritten/forgotten in next subset/batch that contains totally different equations/samples. Increasing the number of epochs will not solve this problem.

**Doomed test cases.** As forgetting is training-related, in testing we focus on a problem we call "doomed" test cases. With the progress on better estimating confidence of a prediction (please refer to Related work section and Methodology section for details), often we know a prediction is very likely to be wrong due to its low confidence. However, since a DNN model is fixed during testing, nothing can be done, and running the same model one more time will produce the same low-confidence prediction. There is no obvious choice if trying to change prediction as other output nodes will have even lower softmax/confidence.

We think that the two problems above are related and can be addressed by incorporating human-like learning characteristics to train and test DNNs. In human learning, once a skill (e.g., recognize the horse in George Stubbs's famous Whistlejacket painting) is acquired, it may be revised for improvement, but it is unlikely to be overwritten/forgotten in order to accommodate a new skill. Instead, skills are accumulated into a skill set. To achieve this in DNNs, one possibility is not to use all available learning capacity and resource (e.g., neurons and connections in a neural network) each time to learn a skill (e.g., correctly classify a sample). Learning of a sample ideally should use a just-enough subset of neurons and connections, not all available ones. When a new sample arrives and can not be correctly predicted with current skill (neurons and connections) set, a new subset of neurons and connections in reserve should be taken in and trained to correctly classify this new sample and saved as a new skill in the skill set. open-style learning, where training becomes a process of building a skill set, not producing a fixed/unchangable model. Accordingly testing procedure will become a dynamic process of choosing suitable skills to confidently classify a specific test sample. If one selection of skills does not reach the confidence threshold, a different subset of skills should be chosen to improve prediction confidence to the threshold or until choices are exhausted. More specifically, this paper proposes a new open-style deep learning model called Supermodel, which is different from standard DNN modeling process in the following aspects:

- During training, useful groups and patterns of nodes and their connections will be collected and saved into a "super" network called Supermodel, which will be treated as a knowledge base or skill set.
- During testing, a test sample will be input to Supermodel and trigger/activate certain neurons and connections. When a prediction is made, its confidence is assessed. If the confidence is above a threshold, we output the prediction. If not, we propose a static masking method and a dynamic masking method to actively select suitable skills (e.g., neurons and connections) and try to improve prediction confidence.

With this open-style learning, our contributions are:

- Skill proposed in this paper is a new component for a DNN besides commonly used architectural components (e.g., layer, node), and may offer new insights into various aspects of DNN modeling.
- Most DNN-based ensemble methods do not break model boundary and merge networks. Skill directly connected to model performance (not like a layer or node) is more versatile and can be selected case-by-case to achieve optimal outcome.
- Besides outperforming SOTA ensemble methods, using commonly available base models Supermodel performs well comparing with the best systems on leaderboard.

In next section, we will present related work. Then in the Methodology section, we will discuss our new training and testing procedure with more details on confidence and two skill selection/masking methods. We evaluated our approach with multiple datasets, backbones, and various settings in Experiment section, and conclude in the last section.

## 2 RELATED WORKS

**Continual learning and catastrophic forgetting.** A key challenge for general AI is how to accommodate learning of a large sequence of tasks involving different objectives, datasets, and modalities. With current DNN modeling scheme, catastrophic forgetting occurs when weights in a DNN well-trained for one task are updated/forgotten during training on data from a new task. Kirkpatrick et al. (2017) proposed to slow down the updating of weights important to previous tasks to reduce forgetting. Sarnen et al. (2020) combines Adversarial Direction with Elastic Weight Consolidation. Similar ideas have been applied to GNN Liu et al. (2021) and Reinforcement Learning Gai et al. (2024) recently. However, current methods that rely on identifying important parameters and slowing down their updates often face conflicts because the closed-style modeling process reuses the same parameters across tasks when old and new tasks require contradictory parameter updates. Expansion-based methods Veniat et al. (2021); Yu et al. (2024); Mendez & EATON (2021); Yoon et al. (2023) address this by adding new nodes and modules during training time when a domain shift or new task is detected. However, how to select suitable parts of a model to deal with a task remains an open challenge. Instead, we propose an open-style method, training separate DNNs to cover distinct subsets of data and merging them into a Supermodel, which only keeps distinct features and patterns in a skill base.

**Ensemble learning.** Traditional ensemble methods integrate predictions of multiple models directly to improve overall performance via bagging, boosting, stacking, Bayesian model averaging and combination Opitz & Maclin (1999); Ganaie et al. (2022). In deep learning, recent ensemble methods Mohammed & Kora (2022)Dang et al. (2025)Li et al. (2021)Benton et al. (2021) started to break down model boundaries and merge multiple DNNs into one DNN model through some direct process such as weight averaging Zou et al. (2021); Shin et al. (2021), Hadamard product of weight matrices Wen et al. (2020)von Oswald et al. (2021), which inevitably leads to loss or damaging of useful patterns/features obtained in training. In contrast, our Supermodel saves all useful patterns and adaptively selects suitable ones during testing for optimal prediction.

## 3 METHODOLOGY

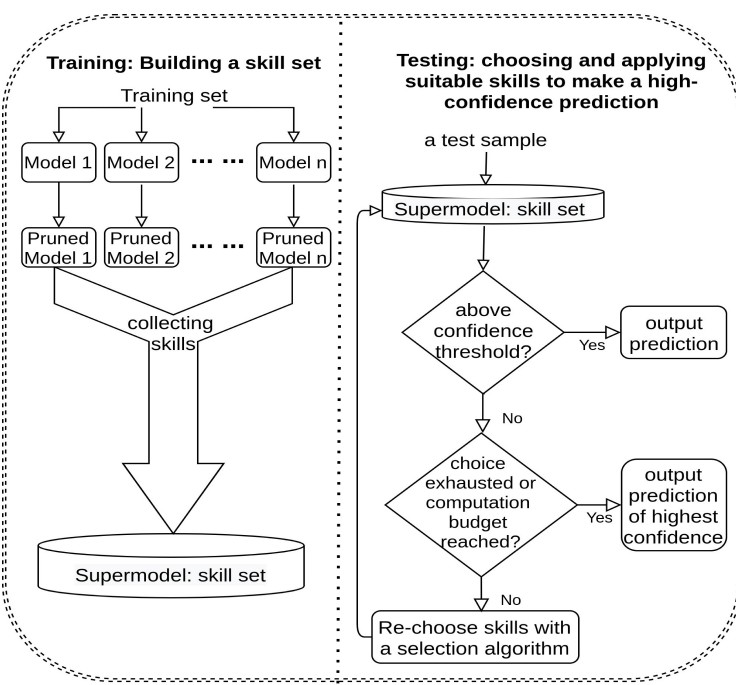

Figure 1: Supermodel architecture: training phase and testing phase in our open-style deep learning model

As shown in Figure 1, our deep learning architecture Supermodel includes a training phase that collects useful skills into a skill set/knowledge base, and a testing phase that selects and applies suitable skills on a test sample to improve its prediction confidence until a threshold is reached or choice/computation budget is exhausted.

## 3.1 TRAINING: COLLECTING A DIVERSE SKILL SET

The goal of training phase in the Supermodel deep learning architecture is to collect a large amount of diverse skills (see Definition 1). We define a skill as:

**Definition 1 (Skill)** *In a neural network $N$, a skill $s$ consists of a node $n \in N$ and all $n$'s incoming connections $\{c_n\}$.*

Now following Figure 1, we will discuss the 3 steps in the training procedure: obtaining multiple models, pruning, and collecting distinct skills to build Supermodel.

### 3.1.1 BUILDING DIVERSE BASE MODELS

Model diversity is important for ensemble learning Kunapuli (2023).Similarly in Supermodel, we need collect distinct skills from a set of diverse base models. Here is the model building procedure: (1) Standard DNN training to obtain the first base model; (2) Recursively use incorrect training samples to train and obtain next model. During each iteration, if a skill is found to have already existed in previous model, weights in this skill will be reset to force generation of new distinct skills; (3) iterate until the number of incorrect samples is less than a threshold or computation budget is exhausted. Multiple diverse models can be effectively built in this way. Details and pseudocode for this procedure is provided in Appendix.

One general concern with ensemble learning is the high computation cost to obtain multiple models. We give a detailed analysis on time complexity in Appendix. Empirically we found our approach is not more expensive than other ensemble learning methods (e.g., requiring a large number of base models). Moreover, Supermodel should be viewed as a knowledge base (not like a single-use model) that can be shared and reused by many tasks and applications, so the higher computation cost is well justified.

### 3.1.2 PRUNING.

A DNN often has a lot of redundancy. Many pruning methods have been developed, which can usually achieve 80%-90% pruning ratio with only a minor drop on model performance. Pruning can help remove trivial and unimportant skills. Moreover, since we want to collect distinct skills, skills with sparse connections are easier to compare and save only new skills into Supermodel. In experiments we choose GRASP pruning method Wang et al. (2020).

### 3.1.3 COLLECTING DISTINCT SKILLS TO BUILD SUPERMODEL

With $N$ pruned models obtained in last step, we compare skills in each layer of each model to skills of corresponding layer in current Supermodel. Two skills are considered similar if their incoming connections come from the same lower-layer nodes and the weight difference between each pair of corresponding connections is less than a threshold $\tau$. Distinct skills are added into Supermodel. It is worth noting that skills in convolution layers can be compared and added in the same way as skills in full-connection layers in Supermodel with the parameter sharing scheme. For skills in convolution layers, we consider the accumulated score of connections sharing the same weight and examine them just like regular connections. A detailed Algorithm with the procedure to collect distinct skills from pruned base models and build a Supermodel skill set is presented in the appendix (Algorithm3).

## 3.2 TESTING: CONFIDENCE-GUIDED PREDICTION ASSESSING AND SKILL SELECTING

During testing, since no class labels are available, Supermodel depends on confidence of a prediction to determine whether currently applied skills work well (e.g., above a confidence threshold) on a test sample. If not, Supermodel will re-select skills to improve the prediction confidence. We will discuss various confidence measures first, then present two skill selection algorithms.

### 3.2.1 CONFIDENCE

Estimation of prediction confidence is useful to many ML applications Pearce et al. (2021). Several measures can be produced directly from softmax layer, such as class probability, negative entropy, and margin . Softmax uncertainty measures may be overconfident or even arbitrary Mukhoti et al. (2021) because of its inability to estimate epistemic uncertainty, and only achieve modest success on OOD tests Pearce et al. (2021). In the experiments since data is mostly IID, we adopt softmax as the confidence measure due to its accuracy on IID data and computation efficiency. Several more reliable measures have been proposed recently, such as calibrated confidence combining density and softmax entropy Mukhoti et al. (2021), correctness ranking loss Mukhoti et al. (2021), which can be adopted in OOD cases.

### 3.2.2 STATIC SKILL SELECTION

---

**Algorithm 1** Static skill selection in testing phase

---

**Require:** Supermodel with weights $W$, number of classes $M$, testing samples $X$, confidence threshold $\alpha$
**Ensure:** final classification result $Y$
1: **for** i from 1 to M **do**
2:     $Mask_i \leftarrow J$. Here $J$ is an all-one matrix of the same size as $W$
3:     **for** all samples in class $i$ **do**
4:         $m \leftarrow \partial conf_i / \partial W$
5:         **for** each element $j$ in $m$ **do**
6:             **if** $j \leq 0$ **then**
7:                 $Mask_{ij} \leftarrow 0$
8:             **end if**
9:         **end for**
10:     **end for**
11: **end for**
12: **for** each $x_i$ in $X$ **do**
13:     **if** max confidence $conf > \alpha$ **then**
14:         $y_i \leftarrow$ class label with max confidence
15:     **else**
16:         $D \leftarrow$ descending sort confidence class indices
17:         **for** each class $j$ in $D$ **do**
18:             recompute confidence of class j $Conf'_j$ with $W \leftarrow W \odot Mask_j$. Here $\odot$ is element-wise multiplication.
19:             **if** $conf'_j > \alpha$ **then**
20:                 $y_i \leftarrow j$
21:                 break
22:             **end if**
23:         **end for**
24:     **end if**
25:     **if** $y_i$ is not assigned **then**
26:         $y_i \leftarrow$ class label with max confidence in $Conf'$
27:     **end if**
28: **end for**

---

Algorithm 1 presents our static skill selection algorithm, where "static" refers to producing a mask for each class during training time. A mask is a binary matrix of the same size as model weight matrix and is applied on the level of individual weights (e.g., indicate that a weight is on or off with 1 or 0). Mask matrix is initialized with all $1's$. Then for each training sample in a class, we calculate the partial derivative of confidence on each weight in Supermodel. If derivative is 0 or negative, we change the mask value corresponding to this weight to 0, so this weight will not be used in the mask for this class. In this way, we generate a static mask for each class. During testing phase, if a testing sample generates a confident prediction, we output this prediction directly. Otherwise, we apply masks of classes with high confidences. If one of these class masks improves the prediction

for this sample above a confidence threshold, we output this prediction. Otherwise, we output the prediction with the highest confidence.

### 3.2.3 DYNAMIC SKILL SELECTION

---
**Algorithm 2** Dynamic skill selection in testing phase

---
**Require:** Supermodel with weights $W$, number of classes $M$, testing samples $X$, confidence threshold $\alpha$, backpropagation threshold $\beta$
**Ensure:** final classification result $Y$
 1: **for** each $X_i$ in $X$ **do**
 2:    **if** max confidence $conf > \alpha$ **then**
 3:        $Y_i \leftarrow$ max confidence class index
 4:        $Mem \leftarrow Y_i$
 5:    **else**
 6:        $count \leftarrow 0$
 7:        **while** max confidence $conf \leq \alpha$ and $count < \beta$ **do**
 8:            $conf_s = Smooth - Max(conf)$
 9:            backpropagate and update $W$ with $conf_s$ as loss
 10:           recompute max confidence $conf$
 11:           $count \leftarrow count + 1$
 12:        **end while**
 13:        **if** max confidence $conf > \alpha$ **then**
 14:           $Y_i \leftarrow$ max confidence class index
 15:        **else**
 16:           $Y_i \leftarrow Mem$
 17:        **end if**
 18:    **end if**
 19: **end for**

---

Static masks are generated during training phase, which does not incur extra cost in testing phase. However, it lacks of flexibility needed for challenging test samples. Hence we further develop a dynamic procedure that selects skills during testing phase. Since labels are not available during testing, we use confidence to replace loss in an objective function and try to maximize:

$$max(c_1, ..., c_n) \tag{1}$$

where $c_i$ is the confidence of each output node for a test sample. The objective is to maximize the max confidence. Since max function is not differentiable, we maximize its smooth approximation instead Lange et al. (2014):

$$Smooth - Max(c_1, ..., c_n) = \frac{\sum_{i=1}^{n} c_i e^{c_i}}{\sum_{i=1}^{n} e^{c_i}} \tag{2}$$

Other options exist, such as maximizing entropy of confidence values. Standard backpropagation is applied until confidence of the most confident output node is above a threshold or computation budget is exhausted. In this way we try to select the best skills we own to make a prediction with as high confidence as possible for otherwise "doomed"/difficult test samples.

## 4 EXPERIMENTS

Since this paper focuses on one task/dataset, we choose the following SOTA ensemble methods for comparison: Snapshot Ensemble (SSE) Huang et al. (2017), Fast Geometric Ensembling (FGE) Garipov et al. (2018), Diversified Ensemble Neural Network (DEns) Zhang et al. (2020), Collegial Ensembles (CE) Littwin et al. (2020), MC-Dropout(MC-Drop) Gal & Ghahramani (2016), BatchEnsemble (Batch-E) , NativeEnsemble (Native-E) Wen et al. (2020). For a comprehensive evaluation, three backbone DNNs (VGG16, WRN-28-10, ResNet-164) and three commonly used

| Backbone | Model | Cifar100 Accu. | Improvement over base model | Cifar10 Accu. | Improvement over base model |
|---|---|---|---|---|---|
| VGG16 | Base model | 72.6 | 0 | 93.25 | 0 |
| | SSE | 73.6 | 1.00 | 93.43 | 0.18 |
| | FGE | 74.3 | 1.70 | 93.52 | 0.27 |
| | DEns | 70.48 | -2.12 | 93.56 | 0.31 |
| | Supermodel | 89.33 | 16.73 | 94.33 | 1.08 |
| | Supermodel-s | 89.84 | 17.24 | 94.36 | 1.11 |
| | Supermodel-d | **90.06** | **17.46** | **94.39** | **1.14** |
| WRN-28-10 | Base model | 80.8 | 0 | 96.18 | 0 |
| | SSE | 82.1 | 1.30 | 96.27 | 0.09 |
| | FGE | 82.3 | 1.50 | 96.35 | 0.17 |
| | CE-101×10d | 82.56 | 1.76 | 96.35 | 0.17 |
| | CE-3×58d | 81.25 | 0.45 | 96.13 | -0.05 |
| | CE-2×98d | 81.68 | 0.88 | 96.01 | -0.17 |
| | Supermodel | 88.75 | 7.95 | 96.44 | 0.26 |
| | Supermodel-s | **89.98** | **9.18** | 96.72 | 0.54 |
| | Supermodel-d | 89.95 | 9.15 | **96.79** | **0.61** |
| ResNet-164 | Base model | 78.5 | 0 | 95.28 | 0 |
| | SSE | 79.1 | 0.60 | 95.34 | 0.06 |
| | FGE | 79.8 | 1.30 | 95.46 | 0.18 |
| | Supermodel | 88.93 | 10.43 | 96.33 | 1.05 |
| | Supermodel-s | 89.40 | 10.90 | 96.66 | 1.38 |
| | Supermodel-d | **89.48** | **10.98** | **96.71** | **1.43** |

Table 1: Experiment results with three backbones: VGG16, WRN-28-10, ResNet-164 on CIFAR10 and CIFAR100. Results on SSE, FGE, DEns, CE, MC-Drop, Batch-E, Native-E are taken from the original papers respectively.

benchmark datasets (CIFAR10, CIFAR100, ImageNet) are selected. Detailed settings including hyperparameters are provided in Appendix.

## 4.1 RESULTS

Table 1 shows the main experiment results using three backbone networks: VGG16, WRN-28-10, ResNet-164 on CIFAR10 and CIFAR100 datasets comparing with SOTA ensemble learning methods. We include three variations of our method: (1) Supermodel: use the constructed Supermodel directly as a regular DNN; (2) Supermodel-s: apply static mask algorithm presented in Algorithm 1 on Supermodel to statically select suitable skills; (3) Supermodel-d: apply dynamic mask algorithm presented in Algorithm 2 on Supermodel to dynamically select suitable skills. All three variations of Supermodel significantly improves the performance over baseline models and outperforms the existing SOTA models. One interesting finding is that Supermodel can make up the constraints of learning capacity in base models. For example, VGG19 and ResNet-164 base models achieves lower accuracy than WRN-28-10, which may indicate that they contain less effective skills, but the performance difference of a single model can be eliminated by skills collected from multiple diverse models.

## 4.2 DISCUSSION

### 4.2.1 COMPARISON WITH A DNN OF THE SAME WIDTH AND NUMBER OF PARAMETERS.

Since supermodel collects distinct skills (e.g., nodes and connections) from multiple base models, one natural question is:

*Is the performance improvement due to a wider network with more parameters?*

| Method | Sparsity | Accuracy |
|---|---|---|
| Base model | 100 | 88.77 |
| Pruned base model | 14.6 | 87.53 |
| Supermodel-d | 14.6 | 89.98 |

Table 2: Comparison with WRN-28-10 of the same width and number of parameters.

| Method | Base model accuracy | Accuracy | Improvement |
|---|---|---|---|
| **ResNet-50 on ImageNet** | | | |
| SSE | 76.13 | 76.67 | 0.54 |
| FGE | 76.13 | 76.69 | 0.56 |
| Supermodel | 78.15 | 85.94 | 7.79 |
| **WRN-28-10 on CIFAR100** | | | |
| Supermodel | 87.26 | 92.93 | 5.67 |

Table 3: Results on ImageNet to assess scalability and impact of stronger base models.

We built a fully connected WRN-28-10 ("Base model" in Table 2) of the same width (e.g., same number of nodes) as our Supermodel, and tested its accuracy on CIFAR100. Then we prune connections in this base model to 14.6%, which is at the same sparsity level as our Supermodel. Supermodel outperforms both models as shown in Table 2 even though the base model contains around 7 times more connections.

#### 4.2.2 DETAILED PERFORMANCE CHANGES IN SKILL COLLECTION PROCESS

As Supermodel construction process includes multiple iterations of model training and skill collection, Table 4 shows performance changes in each iteration. For WRN-28-10, it takes 6 iterations for training accuracy to reach 100%, that is, all training samples are correctly classified by at least one of the 6 models trained in these 6 iterations. Accuracy union is the union of accuracy of these 6 models on testing set. As long as a testing sample can be correctly classified by at least 1 of these 6 models, it will be counted in the union accuracy. Accuracy union can be considered as an upper bound for an ensemble learning method. At each iteration, as more skills are collected, the performance of Supermodel consistently improves as well.

| WRN-28-10 | | | | | |
|---|---|---|---|---|---|
| # | Training accuracy | Accu. union | Super model | Super model-s | Super model-d |
| 1 | 0.8542 | 0.8079 | 0.7924 | 0.8029 | 0.8089 |
| 2 | 0.7462 | 0.8417 | 0.8290 | 0.8351 | 0.8425 |
| 3 | 0.6122 | 0.8882 | 0.8750 | 0.8819 | 0.8843 |
| 4 | 0.4276 | 0.9015 | 0.8835 | 0.8905 | 0.8915 |
| 5 | 0.9724 | 0.9073 | 0.8914 | 0.8943 | 0.8957 |
| 6 | 1.0000 | 0.9118 | 0.8975 | 0.8998 | 0.8983 |
| **ResNet-164** | | | | | |
| # | Training accuracy | Accu. union | Super model | Super model-s | Super model-d |
| 1 | 0.8427 | 0.7885 | 0.7724 | 0.7785 | 0.7715 |
| 2 | 0.8819 | 0.8853 | 0.8628 | 0.8703 | 0.8722 |
| 3 | 1.0000 | 0.9025 | 0.8893 | 0.8940 | 0.8939 |
| **VGG16** | | | | | |
| # | Training accuracy | Accu. union | Super model | Super model-s | Super model-d |
| 1 | 0.7822 | 0.7264 | 0.7139 | 0.7224 | 0.7214 |
| 2 | 0.6214 | 0.8425 | 0.8334 | 0.8395 | 0.8417 |
| 3 | 0.8917 | 0.8824 | 0.8756 | 0.8761 | 0.8792 |
| 4 | 1.0000 | 0.9066 | 0.8933 | 0.8984 | 0.8999 |

Table 4: Detailed performance changes in skill collection process on CIFAR100.

#### 4.2.3 SCALABILITY ASSESSMENT WITH IMAGENET

One ultimate goal of a classifier is the performance. Although Supermodel outperforms SOTA ensemble methods, to evaluate scalability of Supermodel and its performance on large datasets,

we conducted an experiment on ImageNet using ResNet-50 as base model. As shown in Table 3, Supermodel outperforms SSE and FGE, and achieves 85.94% accuracy.

Furthermore to assess the impact of base models, we choose a base model introduced by Ramé et al. (2021) which is trained with 3 original WRN-28-10 model with extra mixing mechanism on samples. This stronger WRN-28-10 base model achieves 87.26% accuracy on CIFAR100. Detailed performance changes in skill collection process are provided in a table in Appendix. With the building procedure described above, Supermodel with dynamic selection achieves 92.93% accuracy, which ranks at the 8th place according to CIFAR100 leaderboard https://paperswithcode.com/sota/image-classification-on-cifar-100 and is 3.09% below the best performing model EffNet-L2. With these results it is reasonable to assume that Supermodel has significant potential for further improvement by adopting even stronger base models.

### 4.2.4 EXPERIMENT WITH NAS METHOD

As discussed above, different from existing ensemble methods, Supermodel does generate a new stand-alone model, which is the objective of a Neural Architecture Search (NAS) approach. However, Supermodel is agnostic and applies to pre-designed DNN models as well as models generated from NAS methods. In this section we adopted a DNN model generated by a SOTA NAS method - DARTS Liu et al. (2018) as a base model. The experiment results in Table 5 show that Supermodel achieves consistent improvement with an already-strong model generated by the SOTA NAS method DARTS.

| Backbone | Method | Accuracy |
|---|---|---|
| DARTS (1st order) | baseline | 0.9692 |
| | supermodel | 0.9728 |
| | supermodel-s | 0.9733 |
| | supermodel-d | 0.9732 |
| DARTS (2nd order) | baseline | 0.9725 |
| | supermodel | 0.9744 |
| | supermodel-s | 0.9745 |
| | supermodel-d | 0.9749 |

Table 5: Results with DARTS on CIFAR10.

| Backbone | Model | Accuracy |
|---|---|---|
| WRN-28-10 | Supermodel-10 | 95.54 |
| | Supermodel-10adjust | 96.22 |
| | Supermodel | 96.79 |

Table 6: Impact of diversity of base models on CIFAR-10.

### 4.2.5 IMPACT OF DIVERSITY OF BASE MODELS.

A set of diverse base models are important for ensemble learning methods. We conducted an experiment with three ways to generate base models. As shown in Table 6, Supermodel-10 combines 10 randomly initialized WRN-28-10 models trained with standard SGD process. Following Garipov et al. (2018) using varied learning rate in training, we train 10 separate models, and Supermodel-10adjust combines these models. Supermodel uses our standard training procedure described in previous section. Results in Table 5 show that Supermodel is stable and not sensitive to base models obtained with different training schemes, which indicates its broad potential application in practice.

## 5 CONCLUSION

In this paper we present a different point of view on DNN training and testing. In our proposed approach Supermodel, DNN training takes an open style and is conducted as a process to acquire diverse skills and build a skill set called Supermodel. During testing, suitable skills are selected statically according to class-based masks or dynamically with backpropagation to produce a high-confidence prediction. With experiments on multiple backbones and datasets, Supermodel outperforms current SOTA methods.

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

# 6 APPENDIX

---

**Algorithm 3** Training procedure to collect distinct skills and build Supermodel skill set

---

**Require:** $N$ pruned models $A_1, A_2, ..., A_N$, network depth $d$, skill similarity threshold $\tau$
**Ensure:** $Supermodel\ Net$
1: assign each input node a unique index, $I_0$ is the set of node indices for input layer 0
2: $l \leftarrow 1$
3: $Net \leftarrow$ an empty neural network with the same number of input nodes as a pruned model. All pruned models have the same number of input nodes since they work on the same dataset.
4: **while** $l \leq d$ **do**
5:     **for** each skill $s_i$ in layer $l$ of each pruned model **do**
6:        assign a unique index $E_i = \{\{c_{i1}, c_{i2}, ..., c_{ik}\}, \{w_{i1}, w_{i2}, ..., w_{ik}\}\}$ to $s_i$, where $\{c_{i1}, c_{i2}, ..., c_{ik}\} \in (I_{l-1})^k$ and $\{w_{i1}, w_{i2}, ..., w_{ik}\}$ is the weights of incoming connections $\{c_{i1}, c_{i2}, ..., c_{ik}\}$
7:        **if** $\{c_{i1}, c_{i2}, ..., c_{ik}\}$ come from the same nodes in layer $l-1$ as incoming connections of a skill in $Net$ layer $l$, and $\forall j$ the difference between each pair of corresponding $w_{ij}$ is less than $\tau w_{ij}$ **then**
8:           $s_i$ is discarded
9:        **else**
10:           $s_i$ is added to $Net$ layer $l$
11:        **end if**
12:     **end for**
13:     $l \leftarrow l + 1$
14: **end while**

---

