# OpenReview forum: "Supermodel: Rethinking DNN Training and Testing with Open-style Skill Acquisition and Dynamic Inference"
_ICLR.cc/2026/Conference — ICLR 2026 Conference Withdrawn Submission_

### Official Review · Reviewer_EE8a · 2025-10-23

**Soundness:** 2
**Presentation:** 2
**Contribution:** 1
**Rating:** 2
**Confidence:** 5

**Summary:**

The main motivation of this paper is to alleviate the forgetting of learned skills and doomed test cases in deep neural network (DNN) training and testing. To this end, the paper proposes a supermodel, constructed by collecting useful groups and patterns of nodes and connections, termed a skill set, for super networks during training. During testing, when a prediction’s confidence falls below a threshold, the model employs static masking and dynamic masking methods to select suitable skills and reattempt the prediction to reassess confidence. The proposed method is validated on several datasets; however, the comparative experiments are limited and insufficient to convincingly demonstrate the superiority of the proposed method.

**Strengths:**

- The motivation of this paper is interesting. The concepts of “forgetting of learned skills” and “doomed test cases” provide a somewhat perspective for analyzing DNN models.
- The paper is very readable and easy to understand. Even though the code is not provided, the implementation details can be inferred from the main descriptions.
- The algorithms are presented in detail, and since some of them are relatively simple, the reader can grasp them directly from the main manuscript.
- The basic experimental results in Table 1, Table 3, and the NAS-based experiments highlight the strengths of the proposed method, though they are somewhat limited in objectively proving its superiority.

**Weaknesses:**

- Although the paper is easy to read, it contains numerous spacing and typographical errors. A thorough revision is necessary. Specifically, line 131 and line 178 lack proper spacing; in addition, references are inconsistently formatted, e.g., in line 042 citations appear as “(reference)”, but later use a different format. Furthermore, some abbreviations (e.g., IID and OOD in lines 223 and 225) are not defined, which may make the paper difficult to understand for readers unfamiliar with the terminology.
- Some undefined symbols are used in the algorithm descriptions, and certain explanations are unnecessarily verbose, describing concepts that can be easily understood from the main manuscript.
- Although Skill is defined in Definition 1, the concept remains somewhat vague; it would be clearer if illustrated with a figure.
- For Doomed test cases, it is difficult to distinguish how this differs from conventional domain adaptation methods. Do they share the same motivation?
- While the motivation is compelling, the proposed method is rather naïve, lacking mathematical justification or detailed explanation to support its effectiveness.
- The performance comparison tables (Tables 1 and 3) include baseline methods from as early as 2017-2018, which limits the ability to convincingly demonstrate the superiority of the proposed method.
- The Related Works section lacks discussion of and comparison with recent studies, making it insufficient to establish the originality of this paper.

**Questions:**

- The result in Table 3 should include an ablation or comparison between Static Masking and Dynamic Masking experiments to validate their respective contributions.

---

### Official Review · Reviewer_31aN · 2025-10-28

**Soundness:** 2
**Presentation:** 2
**Contribution:** 2
**Rating:** 0
**Confidence:** 4

**Summary:**

The paper proposes a method to train an ensemble of neural networks following boosting, followed by pruning of the resulting networks. During training a set of so called skills are determined, which correspond to layers that are different from each other based on the distance between their weight matrices, and a set of static masks are created each of which corresponds to a combination of the so-called skills. The method is compared with ensemble neural network methods on three datasets and shows some advantages.

Close to the proposed method work can be found in the topic of dynamic inference of neural networks, in which different input samples are classified by a traversing a different computational graph of the neural network architecture. However, this line of work is not mentioned and not compared with.

The paper claims that the method solves the open-style skill acquisition problem, however, the experimental process used to evaluate its performance is the standard (so-called by the paper closed-style) classification setup.

**Strengths:**

- The proposed method is compared with some recent ensemble learning methods and provides competitive performance.

**Weaknesses:**

- The claim that the supermodel should be viewed as a knowledge base is unfounded.
 - A line of work which received considerable attention in the last years (dynamic inference) and is directly connected to the proposed method is neglected.
 - Since in general multiple inference steps are performed for each input sample, it is not clear why this information is not used for improving performance as done in ensemble network approaches. For example, the training process looks similar to the (Layer Ensembles 2024), which then exploits the results of different networks in the ensemble to also provide uncertainty estimation of the classification result.
 - The NAS methods used in the comparisons is not SOTA, rather an outdated model from 2018.
 - The main paper text states that "We give a detailed analysis on time complexity in Appendix.", however, the Appendix provides just the algorithm of the training procedure.

**Questions:**

- How does the proposed method compare with dynamic inference approach?
 - What would be the effect of using other metrics for estimating the confidence of the network for each sample?
 - How does the proposed method compare with SOTA NAS methods?
 - What is the time complexity of the method?

---

### Official Review · Reviewer_hy2e · 2025-10-29

**Soundness:** 2
**Presentation:** 2
**Contribution:** 3
**Rating:** 2
**Confidence:** 4

**Summary:**

The paper proposes "Supermodel": a collection of skills (defined as nodes and their incoming connections). During inference the model tries to optimally use a subset of the skills while maximizing the confidence.

**Strengths:**

1. The idea of using a subset of skills during inference is novel and interesting.

2. Figure 1 nicely summarizes the approach.

3. Provides psudocode for all of their algorithms.

**Weaknesses:**

1. The paper starts by proposing forgetting and doomed test case as the core problem to address. I was expecting a continual learning algorithm until I read the first line of Section 4. The paper definitely wrongly motivates the problem and I do not know what they are trying to show in Table 1, 2, 3 and 4. Is it showing mitigation of forgetting or doomed test case? I believe it shows none. What are the baselines? Are they continual learning approaches? Why?

2. Elaborate Line 57-58 on Page 2. What do you mean by "s will become an equation in this system"?

3. My biggest concern is the over-reliance on calibration. The model do not study calibration at all. It claims softmax as good calibration approach whereas literature says otherwise [1]. There are also 2 kinds of calibration: ID and OOD. It completely ignores them.

4. It vaguely refers to Appendix for computational complexity, hyper parameter settings whereas Appendix only contains pseudocode for Algorithm 3.

5. There are no error bars anywhere in the results.

6. Section 3.1.2: what is GRASP? Write a small description about it.

7. Definition 1: Define correctly. Why is a neural network N. Do you mean the collection of nodes in a network as $\mathcal{N}$?




[1] Dey, J., Xu, H., De Silva, A., & Vogelstein, J. T. Simple Calibration via Geodesic Kernels. Transactions on Machine Learning Research.

**Questions:**

See my points in the weakness.

---

### Official Review · Reviewer_DFs1 · 2025-10-31

**Soundness:** 2
**Presentation:** 2
**Contribution:** 3
**Rating:** 4
**Confidence:** 3

**Summary:**

This paper proposes an ensemble learning method called Supermodel, designed to address the forgetting problem during DNN training and the doomed test cases issue during testing. During training, the method selects the samples misclassified in the previous round as input for the next model, thus building a diverse set of base models. Redundant connections are then removed through pruning. Finally, the connection patterns among neurons from these models are regarded as a collection of skills, forming the Supermodel. During testing, a combination of static masks and dynamic selection is used to choose specific skills for prediction.

**Strengths:**

1. The proposed approach for constructing the Supermodel is quite innovative — it treats different neuron connection patterns in DNNs as a set of skills, which is a unique perspective.
2. The testing strategy using static masking and dynamic adjustment shows certain effectiveness.
3. The paper provides detailed experimental validation and reports the performance at each iteration.

**Weaknesses:**

1. The ensemble learning methods compared with the authors’ Supermodel are relatively outdated — the newest one used is the Dens method from 2020.
2. The dynamic backpropagation used during testing adds extra computational cost, but the performance improvement is not very significant.

**Questions:**

1. Comparison methods: The ensemble methods used for comparison are only up to 2020. Are there any more recent SOTA methods that could be included?
2. Similarity threshold: How is the threshold that determines the similarity between skills set? If it’s too low, redundant skills may appear; if it’s too high, model diversity could be lost.

---

### Note · Authors · 2025-11-12

I have read and agree with the venue's withdrawal policy on behalf of myself and my co-authors.